# Potent SARS-CoV-2 neutralizing antibodies with protective efficacy against newly emerged mutational variants

Tingting Li [1,2,12], Xiaojian Han[1,2,12], Chenjian Gu[3,12], Hangtian Guo [4,5,12], Huajun Zhang [6,12], Yingming Wang [1,2,12], Chao Hu[1,2], Kai Wang [7], Fengjiang Liu [4], Feiyang Luo[1,2], Yanan Zhang[8,9], Jie Hu[7], Wang Wang[1,2], Shenglong Li[1,2], Yanan Hao[1,2], Meiying Shen[10], Jingjing Huang[1,2], Yingyi Long[1,2], Shuyi Song[1,2], Ruixin Wu[1,2], Song Mu[1,2], Qian Chen[1,2], Fengxia Gao[1,2], Jianwei Wang[1,2], Shunhua Long[1,2], Luo Li[1,2], Yang Wu[3], Yan Gao [4], Wei Xu[3], Xia Cai[3], Di Qu[3], Zherui Zhang[8,9], Hongqing Zhang[8,9], Na Li[8,9], Qingzhu Gao[7], Guiji Zhang[7], Changlong He[7], Wei Wang [11], Xiaoyun Ji[5,11], Ni Tang [7], Zhenghong Yuan [3], Youhua Xie [3✉], Haitao Yang [4✉], Bo Zhang [8✉], Ailong Huang [7✉] & Aishun Jin [1,2✉]

Accumulating mutations in the SARS-CoV-2 Spike (S) protein can increase the possibility of immune escape, challenging the present COVID-19 prophylaxis and clinical interventions. Here, 3 receptor binding domain (RBD) specific monoclonal antibodies (mAbs), 58G6, 510A5 and 13G9, with high neutralizing potency blocking authentic SARS-CoV-2 virus display remarkable efficacy against authentic B.1.351 virus. Surprisingly, structural analysis has revealed that 58G6 and 13G9 both recognize the steric region $S^{470-495}$ on the RBD, over-lapping the E484K mutation presented in B.1.351. Also, 58G6 directly binds to another region $S^{450-458}$ in the RBD. Significantly, 58G6 and 510A5 both demonstrate prophylactic efficacy against authentic SARS-CoV-2 and B.1.351 viruses in the transgenic mice expressing human ACE2 (hACE2), protecting weight loss and reducing virus loads. Together, we have evidenced 2 potent neutralizing Abs with unique mechanism targeting authentic SARS-CoV-2 mutants, which can be promising candidates to fulfill the urgent needs for the prolonged COVID-19 pandemic.

[1] Department of Immunology, College of Basic Medicine, Chongqing Medical University, Chongqing 400010, China. [2] Chongqing Key Laboratory of Basic and Translational Research of Tumor Immunology, Chongqing Medical University, Chongqing 400010, China. [3] Key Laboratory of Medical Molecular Virology, Department of Medical Microbiology and Parasitology, School of Basic Medical Sciences, Shanghai Medical College, Fudan University, Shanghai 200032, China. [4] Shanghai Institute for Advanced Immunochemical Studies and School of Life Science and Technology, ShanghaiTech University, Shanghai 201210, China. [5] State Key Laboratory of Pharmaceutical Biotechnology, School of Life Sciences, Nanjing University, Nanjing, Jiangsu 210023, China. [6] State Key Laboratory of Virology, Wuhan Institute of Virology, Center for Biosafety Mega-Science, Chinese Academy of Sciences, Wuhan 430071, China. [7] Key Laboratory of Molecular Biology on Infectious Diseases, Ministry of Education, Chongqing Medical University, Chongqing 400010, China. [8] Key Laboratory of Special Pathogens and Biosafety, Wuhan Institute of Virology, Center for Biosafety Mega-Science, Chinese Academy of Sciences, Wuhan 430071, China. [9] University of Chinese Academy of Sciences, Beijing 100049, China. [10] Department of Breast Surgery, Harbin Medical University Cancer Hospital, Harbin 150000, China. [11] Institute of life sciences, Chongqing Medical University, Chongqing 400010, China. [12] These authors contributed equally: Tingting Li, Xiaojian Han, Chenjian Gu, Hangtian Guo, Huajun Zhang, Yingming Wang. ✉email: yhxie@fudan.edu.cn; yanght@shanghaitech.edu.cn; zhangbo@wh.iov.cn; ahuang@cqmu.edu.cn; aishunjin@cqmu.edu.cn

The persistence of COVID-19 in the global population can result in the accumulation of specific mutations of SARS-CoV-2 with increased infectivity and/or reduced susceptibility to neutralization[1–11]. Highly transmissible SARS-CoV-2 variants, such as B.1.351 emerged in South Africa, harbor multiple immune escape mutations, and have raised global concerns for the efficacy of available interventions and for re-infection[2–9,11]. As these challenges presented, the protective efficacy of current antibody-based countermeasures needs to be thoroughly assessed against the current mutational variants.

The major interest of neutralizing therapies has been targeted towards SARS-CoV-2 RBD, which is the core region for the host cell receptor ACE2 engagement[12–22]. B.1.351 bears 3 mutations, $S^{K417N}$, $S^{E484K}$ and $S^{N501Y}$, in its RBD, the first 2 of which have been proven to be the cause for its evasion from neutralizing Ab and serum responses[2–9]. Nevertheless, a small group of SARS-CoV-2 RBD specific neutralizing Abs demonstrated undisturbed in vitro potency against B.1.351[2–7,9]. Evaluating their therapeutic efficacy against the circulating strains is necessary for the reformulation of protective interventions and vaccines against the evolving pandemic.

Here, we have focused on 20 neutralizing Abs selected from a SARS-CoV-2 RBD specific mAb reservoir and confirmed their potency against authentic SARS-CoV-2 virus. Excitingly, at least 3 of our mAbs exhibit remarkable neutralizing efficacy against authentic B.1.351 virus. 58G6, one of our top neutralizing Abs, targets a region of $S^{450-458}$ and a steric site $S^{470-495}$ on the receptor binding motif (RBM). Furthermore, potent 58G6 and 510A5 demonstrate strong prophylactic efficacy in SARS-CoV-2- and B.1.351-infected hACE2-transgenic mice. Collectively, our study has characterized a pair of neutralizing Abs with potential effective therapeutic value in clinical applications, which may provide updated information for RBD specific mAbs against the prolonged COVID-19 pandemic.

## Results

**The neutralizing potency of RBD specific Abs.** By our recently established rapid neutralizing Abs screening system[23], we have successfully obtained 20 neutralizing Abs with high affinities to RBD from COVID-19 convalescent individuals, and their neutralizing potency was confirmed by the half inhibition concentrations ($IC_{50}$s) against authentic SARS-CoV-2 virus quantified via qRT-PCR (Fig. 1a, c and Supplementary Fig. 1). Here, we analyzed the neutralizing potency of our top 10 neutralizing Abs against authentic SARS-CoV-2 and B.1.351 viruses by the plaque-reduction neutralization testing (PRNT). At least 3 of our potent neutralizing Abs 58G6, 510A5 and 13G9 exhibited striking neutralizing efficacy against SARS-CoV-2, with the $IC_{50}$s value ranging from 1.285 to 9.174 ng/mL (Fig. 1b, c). Importantly, the RBD escape mutations of B.1.351 did not compromise the neutralizing efficacy of 58G6 and 510A5 (Fig. 1b, c). As reported for a wide range of RBD specific neutralizing Abs[2–9], authentic B.1.351 virus has challenged some of the tested mAbs (Fig. 1b, c). However, majority of our top 10 mAbs still exhibited neutralizing capabilities against this variant (Fig. 1b, c). Of note, the neutralizing potency of all 10 mAbs against the B.1.1.7 pseudovirus was shown to be similar to those against the SARS-CoV-2 pseudovirus (Fig. 1c and Supplementary Fig. 2). In addition, the binding affinity of 58G6 to the B.1.351 S1 subunit was comparable to that to the SARS-CoV-2 S1, while 510A5 and 13G9 showed higher binding affinity to the S1 subunit of SARS-CoV-2 than that of B.1.351 (Supplementary Fig. 3). Majority of these top 20 neutralizing Abs exhibited no cross-reactivity to the SARS-CoV S protein or the MERS-CoV S protein (Supplementary Fig. 4a). Collectively, 3 RBD specific mAbs demonstrated

potent neutralizing efficacy against authentic SARS-CoV-2 and B.1.351 viruses, suggesting that our neutralizing Abs might be applied for the current COVID-19 pandemic.

**The epitopes characterization for the neutralizing Abs.** To define potential antigenic sites on SARS-CoV-2 RBD, we performed competitive ELISA with the above top 20 neutralizing Abs and the other 54 mAbs selected from our developed RBD-specific mAb reservoir. As shown in Fig. 2a, 5 groups of mAbs were identified according to their recognition sites, each of which consisted of mAbs competing for the epitope for 13G9 (13G9e), the epitope recognized by a non-neutralizing SARS-CoV-2 specific mAb 81A11 (81A11e), or the epitope recognized by a SARS-CoV specific neutralizing Ab CR3022[24] (CR3022e) (Fig. 2a). Interestingly, the epitopes recognized by the majority of potent neutralizing Abs overlapped with 13G9e (Fig. 2a). Next, we confirmed that the top 20 mAbs could directly inhibit the interaction of SARS-CoV-2 RBD and ACE2 by the competitive ELISA and surface plasmon resonance (SPR) assay (Supplementary Figs. 5 and 6). To assess the interrelationships between the epitopes recognized by our top 20 neutralizing Abs in detail, we performed competitive ELISA using biotinylated mAbs. We found that 16 of them competed with 13G9, whereas the antigenic sites of the other 4 Abs (510A5, 55A8, 57F7 and 07C1) overlapped with an independent epitope (510A5e) (Supplementary Fig. 7). These findings suggest that there are at least 2 independent epitopes on the RBD related to SARS-CoV-2 neutralization, from which 13G9e may represent a key antigenic site for the binding of potent neutralizing Abs to the RBD.

To test whether our mAbs could elicit synergistic effect against SARS-CoV-2, we paired each of the top 3 neutralizing Abs (58G6, 510A5 or 13G9) with one Ab exhibiting much lower potency from another group shown in Fig. 2a. Synergistic effects were observed for all combinations at higher levels of inhibition against the authentic virus, confirming the advantage of neutralizing Ab cocktails (Fig. 2b–d).

**A linear binding region in the denatured RBD for 58G6.** To determine the precise interactive regions of our potent neutralizing Abs, first, we assessed the binding ability of the top 20 mAbs to the denatured RBD. In a preliminary screening, 9 mAbs from our top 20 mAbs to SARS-CoV-2 RBD were found to be capable of directly binding to the denatured RBD (Supplementary Fig. 8). Therefore, we designed and synthesized fifteen 20-mer peptides (RBD1 to RBD15), overlapping with 5 amino acids, to cover the entire sequence of RBD, as amino acids 319–541 of SARS-CoV-2 S ($S^{319-541}$) (Supplementary Fig. 9a). Unexpectedly, instead of a continuous linear region, we found that 5 of these 9 mAbs could simultaneously recognize 3 independent fragments (RBD2, RBD9 and RBD13), while 58G6 only strongly bound to RBD9 ($S^{439-459}$) (Supplementary Fig. 9b, c). To determine the essential amino acid residues in the RBD accounted for 58G6 binding, we re-synthesized two 20-mer peptides overlapping with 15 amino acids (RBD9-1 and RBD9-2), covering the RBD9 specific residues (Supplementary Fig. 9a). The results of peptide ELISA revealed that 58G6 preferentially interacted with RBD9-1 than RBD9, in a dose-dependent manner, whereas no interaction of 58G6 with RBD9-2 was observed (Fig. 3a, b). When we individually replaced each amino acid residue in RBD9-1 ($S^{444-463}$) with alanine (A), we found that the binding of 58G6 to a fragment of 8 amino acids ($S^{450-457}$) was markedly reduced (Fig. 3a). To a lesser extent, $S^{445-449}$ and $S^{458-463}$ also slightly affected the binding of 58G6, and the former might explain for the abolished interaction of 58G6 with RBD9-2 (Fig. 3a). Moreover, we found that RBD9-1 bound to ACE2 in a dose-dependent manner, which could be competitively inhibited by 58G6 (Fig. 3c–e). And the

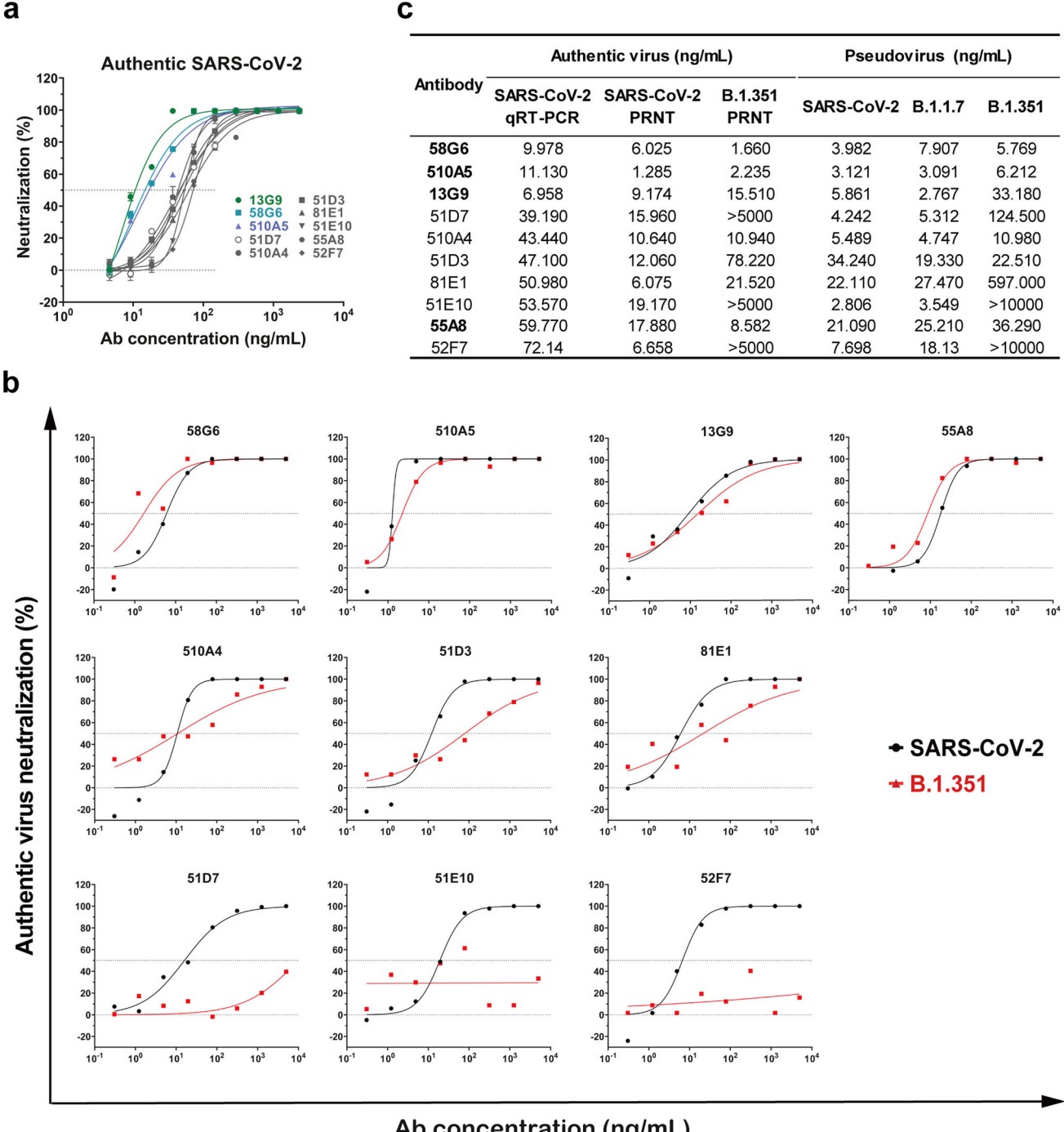

**Fig. 1 The neutralizing capabilities of the top 10 mAbs against authentic SARS-CoV-2 and B.1.351 viruses.** The neutralizing potency of the top 10 mAbs was measured by authentic SARS-CoV-2 (nCoV-SH01) neutralization assay and quantified by **a** qRT-PCR ($n = 3$ biologically independent samples) or authentic SARS-CoV-2 (WIV04) and B.1.351 neutralization assays and quantified by **b** plaque-reduction neutralization testing (PRNT) ($n = 2$ biologically independent samples). Data were presented as **a** mean values ± SEM and **b** mean values. The IC$_{50}$s were summarized in **c**. Dashed line indicated 0% or 50% reduction in viral neutralization. Effective Abs against authentic B.1.351 were shown in bold.

region of S$^{445-463}$ was identified to be critical for the RBD9-1-ACE2 interaction (Fig. 3c, d). Hence, S$^{445-463}$ represents an important region of SARS-CoV-2 RBD for the recognition of neutralizing Abs represented by 58G6. It is worth mentioning that the interaction of 510A5 or 13G9 with the denatured RBD was not observed (Supplementary Fig. 8). Taken together, we evidenced a linear region in the denatured RBD (S$^{450-457}$) that could be recognized by 58G6, which was one of the potent neutralizing Abs against authentic SARS-CoV-2 and B.1.351 viruses.

**The epitopes for 13G9 and 58G6.** To further investigate the molecular mechanism of our neutralizing Abs against SARS-CoV-2, we determined the single-particle cryo-electron microscopy (cryo-EM) structures of the antigen binding fragments (Fabs) of 13G9 or 58G6 in complex with the modified SARS-CoV-2 S trimer with stabilizing mutations[25] (Supplementary Fig. 10a, b). We refined these two complex structures to the overall resolution of 3.9 Å for 13G9 and 3.6 Å for 58G6, respectively (Fig. 4a, b, Supplementary Fig. 10c–j and Supplementary Table. 1). For either the 13G9 or the

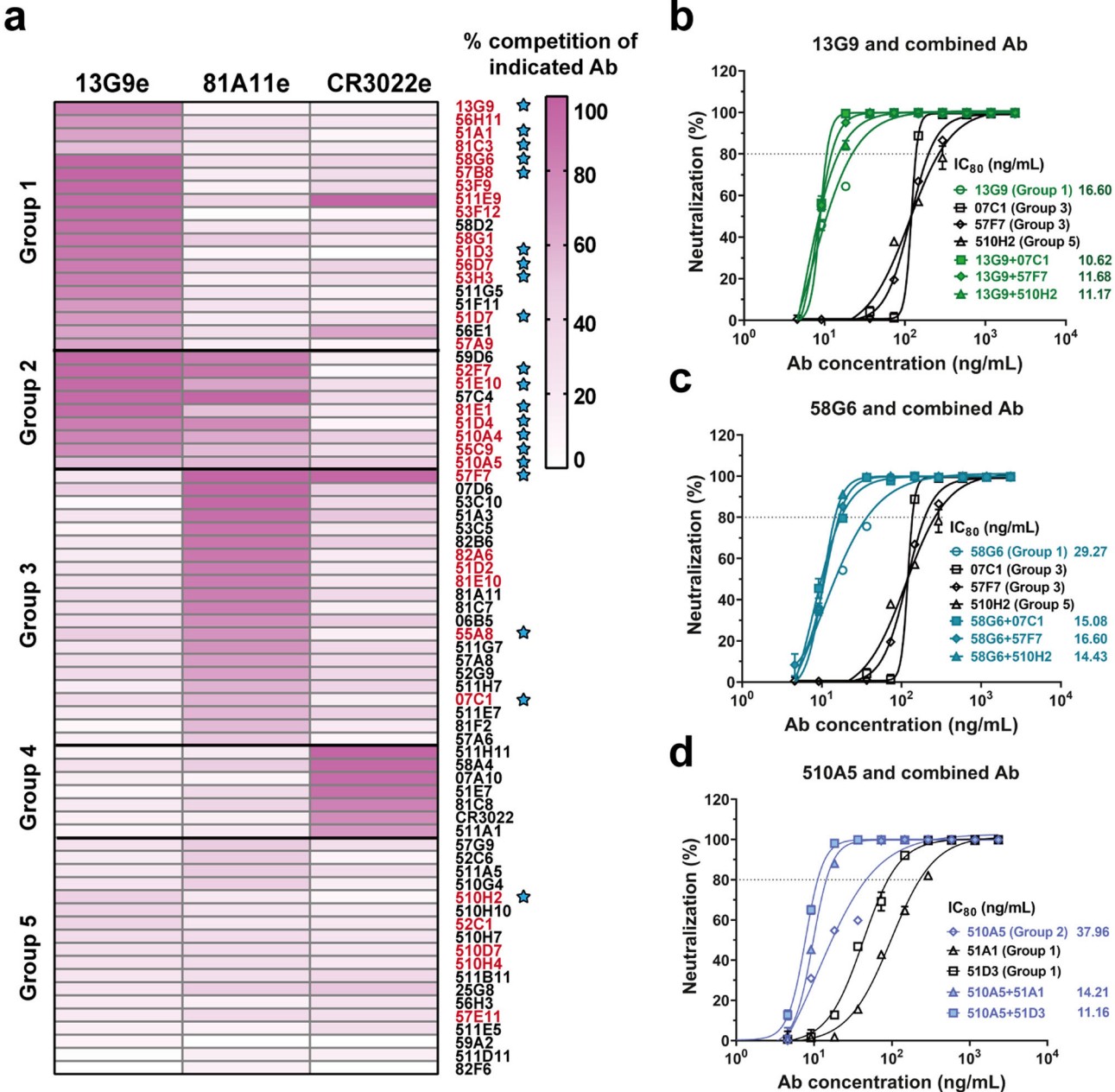

**Fig. 2 Epitope mapping of mAbs and the analysis of neutralizing Abs from different groups. a** Epitope mapping of purified mAbs targeting three independent epitopes (13G9e, 81A11e and CR3022e). All mAbs in Group 1 competed with 13G9; each mAb in Group 3 competed with 81A11; Group 2 consisted of mAbs cross-reacted with 13G9e and 81A11e, the latter to a lesser extent; all mAbs in Group 4 targeted the epitopes overlapping with CR3022e, and the mAbs in Group 5 recognized none of these 3 epitopes. All neutralizing Abs identified by authentic SARS-CoV-2 CPE assay were labeled in red. The top 20 mAbs identified by qRT-PCR with authentic SARS-CoV-2 were indicated by blue stars. The synergistic effects of 13G9 (**b**) and 58G6 (**c**) with 07C1 or 57F7 recognizing 81A11e, or 510H2 with no clearly identified epitope, against authentic SARS-CoV-2 were quantified by qRT-PCR. **d** The synergistic effects of 510A5 with 51A1 or 51D3 recognizing 13G9e, against authentic SARS-CoV-2 were quantified by qRT-PCR. Dashed line indicated 80% inhibition in the viral infectivity. Data for each mAb were obtained from a representative neutralization experiment of three replicates, presented as mean values ± SEM.

58G6 complex, the three-dimensional classification of the cryo-EM data showed the presence of a dominant conformational state of S trimers in complex with the Fabs, with the majority of selected particle images representing a 3-Fab-per-trimer complex (Fig. 4a, b). As shown in Fig. 4a, in individual complex, each 13G9 Fab interacted with one RBD in the "up" state. Similar to the structure of the 13G9 Fab-S complex, only one dominant particle class was observed for the 58G6 Fab-S complex, corresponding to a 3-Fab-bound complex with all 3 RBDs in the "up" conformation (Fig. 4b).

Further refinement of the variable domains of 13G9 or 58G6 and the RBD to 3.8 Å or 3.5 Å, respectively, revealed detailed molecular interactions within their binding interface (Supplementary Fig. 10c±j). These two refined density maps along with the predicted structures of the 13G9 and 58G6 Fabs were used to build the models to illustrate detailed amino acid structures in three dimensions (Supplementary Fig. 11)[26]. Superimposition of the RBDs in the structures of 13G9 Fab-RBD and ACE2-RBD complexes indicates a steric clash between ACE2 and the variable domains on the heavy chain (HC) and the

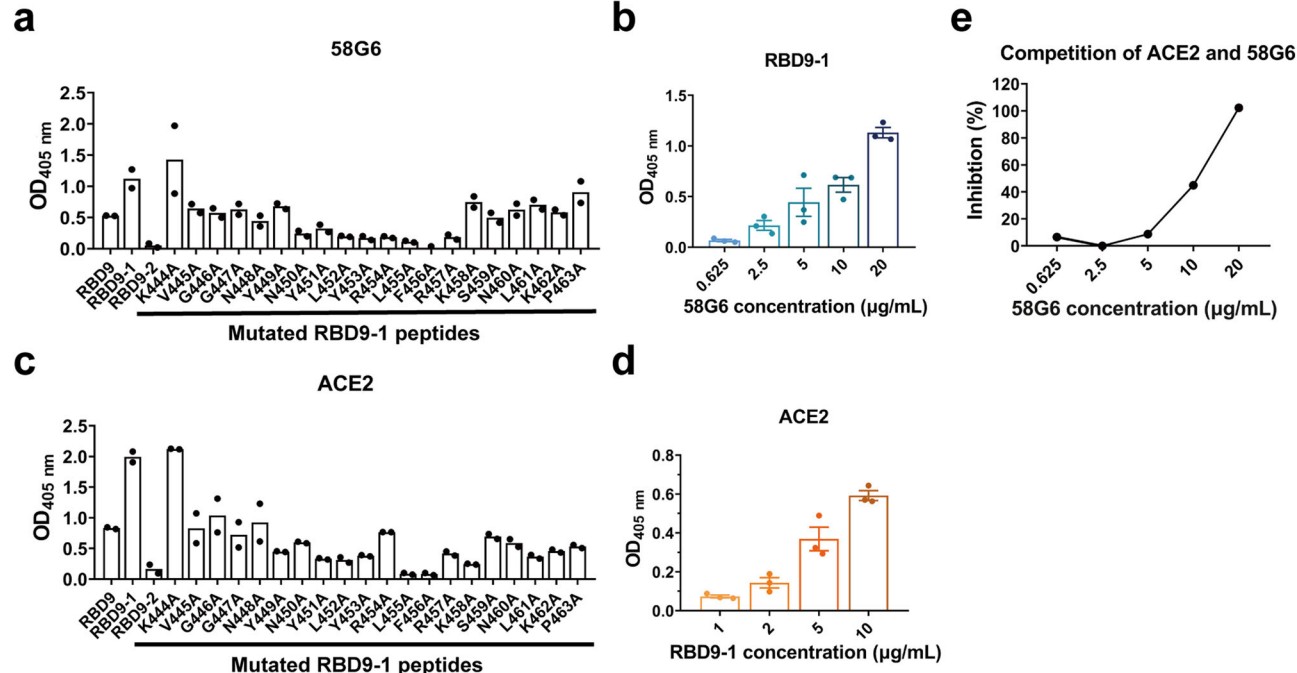

**Fig. 3 The interaction of 58G6 with a linear region in the denatured RBD.** ELISA results of the binding activities of 58G6 (**a**) or ACE2 (**c**) to 3 peptides covering sequences in close proximity, RBD9, RBD9-1 and RBD9-2, and single mutations derived from the full length RDB9-1 ($n = 2$ independent experiments). The binding activity of 58G6 (**b**) or ACE2 (**d**) in various concentrations to the RBD9-1 peptide, tested by ELISA ($n = 3$ independent experiments). Data were presented as mean values ± SEM. **e** The ability of 58G6 in blocking the interaction between RBD9-1 and ACE2, tested by competitive ELISA ($n = 2$ independent experiments).

light chain (LC) of 13G9 Fab (Fig. 4c). Such observations indicate that 13G9 can competitively inhibit the interaction between the SARS-CoV-2 RBD and ACE2. Likewise, an almost identical steric clash between 58G6 Fab and ACE2 was observed, indicating that the SARS-CoV-2 RBD-ACE2 interaction can be prohibited by 58G6 (Fig. 4c). When we compared the details of binding interface of these 2 mAbs and RBD, they showed high level of structural similarity (Fig. 4d).

Specifically, majority of the complementarity determining regions (CDRs; CDRH2, CDRH3, CDRL1 and CDRL3) of 13G9 Fab directly participate in the interaction with the steric region of $S^{470–495}$ (Fig. 5a). Meanwhile, 58G6 Fab was shown to recognize the same steric region using its CDRs: CDRH2, CDRH3, CDRL1 and CDRL3 (Fig. 5b). In parallel, an additional site of residues 450–458 on SARS-CoV-2 S ($S^{450–458}$) was observed for 58G6 recognition (Fig. 5b), which contained the linear region of $S^{450–457}$ we had identified with the denatured RBD, as shown above (Fig. 3a, b).

We found that both 13G9 and 58G6 were derived from IGHV1-58 for the heavy chain and IGKV3-20 for the light chain, with a few differences in amino acid constitution of their CDRH1, CDRH3 and CDRL3 (Supplementary Table. 2). These identical germline gene origins correlated with the structural similarity between 13G9 and 58G6 (Fig. 4d). Several potential hydrogen bonds were identified on the contact surface of each mAb and RBD, representing the unique network associated with individual CDRs and amino acid residues within the epitope corresponding to each mAb (Fig. 5c, d). In summary, these Fab-S complex structures suggest that 13G9 and 58G6 adopt the same potential neutralizing mechanism, wherein they are capable to simultaneously bind to 3 RBDs, occluding the access of SARS-CoV-2 S to ACE2. Notably, R94 in the CDRL3 of 13G9 or N94 in 58G6 forms a hydrogen bond with the carbonyl group on the main chain, rather than the side chain, of $S^{E484}$ (Fig. 5c, d). Moreover, direct contact with a hydrogen bond was found between T105 in the CDRH3 of 58G6 and K458 in the RBD, but not for S105 in 13G9 (Fig. 5c, d).

**The in vivo efficacy of 58G6 and 510A5.** Given the $IC_{50}$s of our mAbs 58G6 and 510A5 against authentic B.1.351 were as low as approximately 2 ng/mL in vitro, we tested their prophylactic efficacy in the transgenic animal model. Different groups of hACE2 mice received intraperitoneal administration of these 2 mAbs or PBS 24 h before an intranasal challenge with authentic SARS-CoV-2 (WIV04) or B.1.351. For the hACE2 mice challenged with SARS-CoV-2 (WIV04), the PBS group showed significant loss of body weight, while those animals from either mAb-treated group retained their body weight for 3 days post-infection (Fig. 6a). When challenged with B.1.351, the hACE2 mice receiving PBS showed gradual weight loss and reached an approximately 30% drop at day 3 (Fig. 6a). Slight weight loss in the hACE2 mice receiving 58G6 or 510A5 treatment was observed at day 1 and day 2. Nevertheless, these 2 mAbs had successfully put the brake on the B.1.351-induced weight reduction by day 3 (Fig. 6a). Importantly, we found that the viral load of either SARS-CoV-2 or B.1.351 in the lung tissues was markedly decreased with a single dose of either mAb (Fig. 6b). These results indicate that 58G6 and 510A5 can effectively protected hACE2 transgenic mice from infectious SARS-CoV-2 and B.1.351, highlighting their prophylactic potential in the present COVID-19 epidemic.

## Discussion

The persistence of COVID-19 has led to generation of mutational variants and immunological adaptation of SARS-CoV-2[1–11]. Newly emerged B.1.351 in South Africa has been reported to confer resistance to neutralization from multiple available mAbs, convalescent plasma and vaccinee sera, posting a high re-infection risk[2–9]. In the present study, we identified 20 neutralizing Abs with high potency against authentic SARS-CoV-2 virus, from a RBD specific mAb reservoir. Among them, 58G6 and 510A5 exhibit high neutralizing capabilities against the authentic virus. Remarkably, these 2 mAbs can efficiently neutralize authentic B.1.351 virus, comparable to most effective

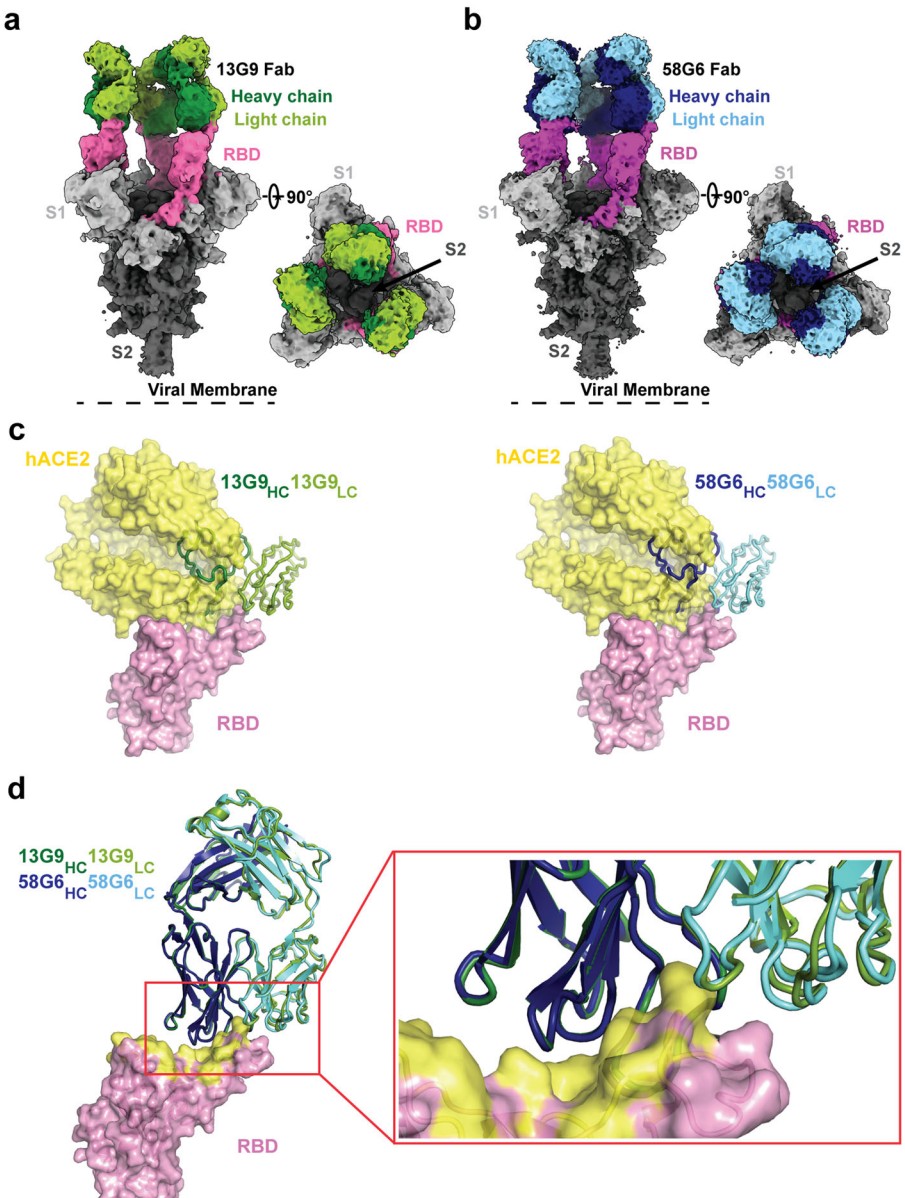

**Fig. 4 Cryo-EM structures of 13G9 and 58G6 Fabs binding to open S trimer. a, b** Cryo-EM densities for 13G9 Fab (antigen binding fragment)-S (a; 3.9 Å) and 58G6 Fab-S (b; 3.6 Å) complexes, revealing binding of 13G9 or 58G6 to RBDs in the all 'up' state. **c** Superimposition of RBD-hACE2 [Protein Data Bank (PDB) ID 6LZG][https://www.rcsb.org/structure/6LZG] complex structure together with RBD-13G9 Fab (left) or RBD-58G6 Fab (right) variable domains, respectively. **d** Alignment of 13G9 and 58G6 Fabs on the same RBD. HC, heavy chain; LC, light chain. CDR, complementarity determining region.

neutralizing Abs reported up to date[2,5,6,9]. Their IC$_{50}$s against this variant were as low as approximately 2 ng/mL, hence we termed these 2 mAbs as potent neutralizing Abs. Such profound neutralizing potency was confirmed in vivo where the prophylactic treatment of these 2 mAbs could efficiently protect the transgenic mice carrying hACE2 against the airway exposure of authentic SARS-CoV-2 and B.1.351 viruses. These results put 58G6 and 510A5 at the center stage for the development of clinically effective therapeutic regiments against the current COVID-19 pandemic. Although the Ab isolation methods (as showing in Methods) we used make it difficult to correlate each specific mAb to the blood sample from which it came from, the discovery of potent neutralizing mAbs such as 58G6 and 510A5 highlighted the possibility of the convalescent patient plasma for the protection against SARS-CoV-2 and its mutants.

In order to understand the high neutralizing potency of our mAbs against SARS-CoV-2, we assessed the antigenic landscape of SARS-CoV-2 RBD. We found that all our RBD targeting mAbs could be categorized into 5 groups according to their recognition on the RBD. Interestingly, the epitopes recognized by the majority of our potent neutralizing Abs overlapped with 13G9e, suggesting that it represented one of the vulnerable sites on SARS-CoV-2 RBD. The other 4 of the top 20 mAbs competed with 510A5 for the binding of RBD at 510A5e. It is worth mentioning that these 2 regions may correspond to 2 separate classes of epitopes recognized by the largest numbers of RBD specific neutralizing Abs, as described in recent studies (Supplementary Fig. 12)[17,27].

In detail, we identified that 58G6 recognized a region consisted of amino acids 450–458 in the RBD. Of note, recent cryo-EM

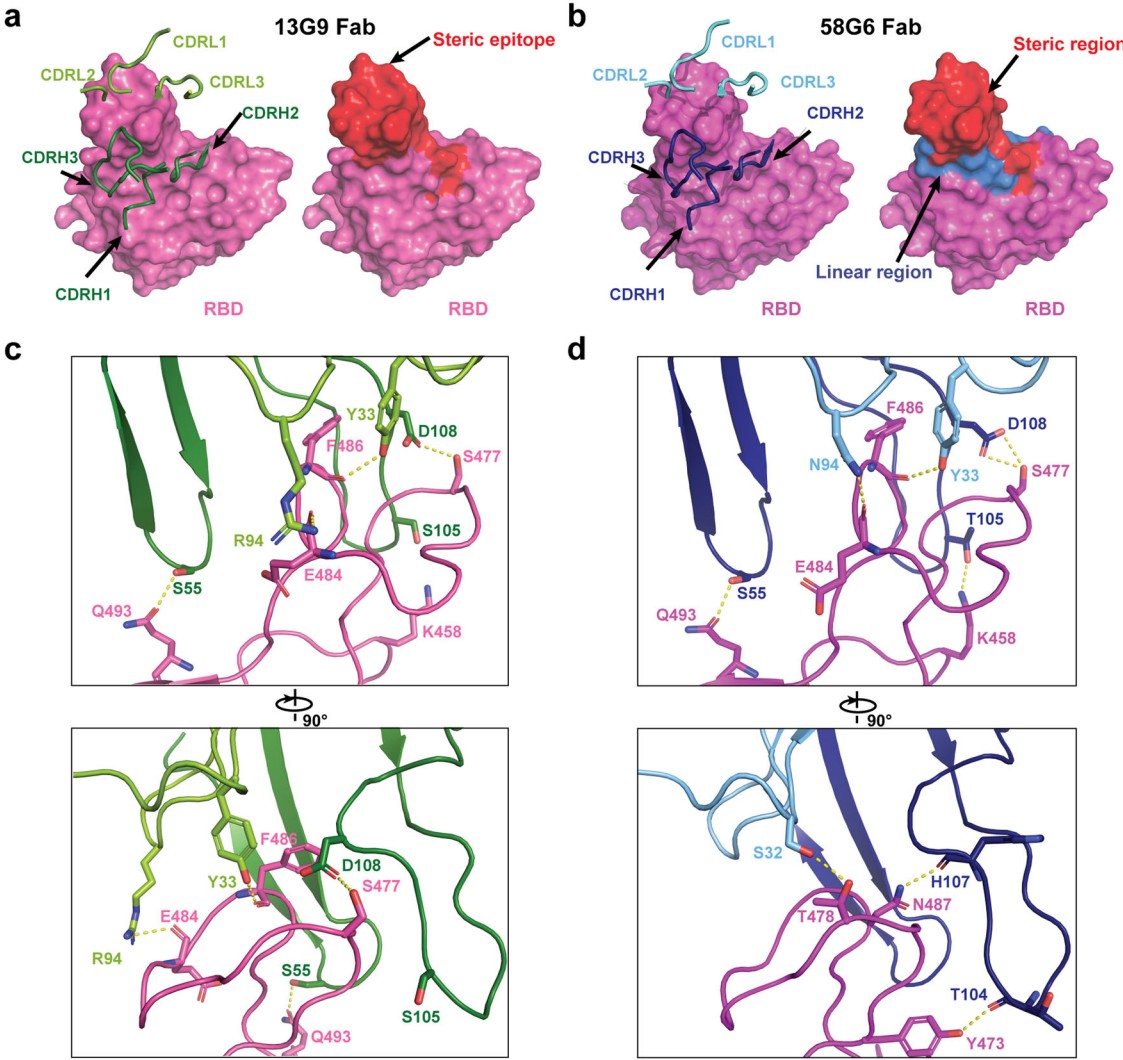

**Fig. 5 Details of interactions between SARS-CoV-2 RBD and mAbs. a**, **b** CDR loops of 13G9 Fab (**a**, left) and 58G6 Fab (**b**, left) overlaid on the surface representation of RBD (shown as pink and magenta, respectively), and surface representations of 13G9 epitope (**a**, right, red) and 58G9 epitope (**b**, right, red and blue) on the RBD surface. **c**, **d** The hydrogen bonds at the binding interface between 13G9 (left) or 58G6 (right) and SARS-CoV-2 RBD.

structure analysis has revealed 3 key ACE2-interacting residues ($S^{Y453}$, $S^{L455}$, and $S^{F456}$)[12,13], indicating that $S^{450-458}$ may be the critical site taken into consideration for SARS-CoV-2 prophylaxis. We found at least one specific hydrogen bond within this region, between 58G6 and RBD, that may contribute to recognition of the unique linear region by 58G6, rather than 13G9. Although certain steric proximity of 13G9 to $S^{450-458}$ has been observed, it needs to be pointed out that no specific linear binding sites have been identified for this mAb.

Moreover, 13G9 and 58G6 both recognized the steric epitope of $S^{470-495}$ on the RBD, which was the key region shared by ACE2 and several reported potent neutralizing Abs against SARS-CoV-2[12,13,21,27]. The cryo-EM analysis revealed a hydrogen bond between N94 in 58G6 and the carbonyl group on the main chain, rather than the side chain, of $S^{E484}$. Common mutation within this region found in current variants, such as $S^{E484K}$ in B.1.351 or P.1 emerged in Brazil[9,11], may not have significant impact on the affinity of 58G6 to the RBDs of these variants. Indeed, the sustained affinity of 58G6 to B.1.351 S1 has been confirmed by the SPR, which may explain for the potentially broad neutralizing spectrum of 58G6. However, for 13G9, the $S^{E484K}$ mutation in B.1.351 or P.1 may introduce an additional positive charge around R94 within its CDRL3, which may lead to strong

electrostatic repulsions between the two residues. This may explain the decreased affinity of 13G9 to B.1.351, hence the slight decrease of neutralizing potency against this variant. To be noted, SARS-CoV presented little similarity in the amino acid sequence corresponding to those accounted for the epitopes recognized by 13G9 or 58G6 on SARS-CoV-2 RBD, and MERS-CoV did not contain such region (Supplementary Fig. 4b). This might be the reason for the lack of cross-reactivity of 58G6 and 13G9 to the other coronaviruses, which was in line with the findings of their binding capability.

For the potent 13G9 or 58G6, we noted that the RBDs interacting with the 3 Fabs of Abs are universally in the 'up' state. As previously described, such full occupancy in each complex could render RBD completely inaccessible for ACE2[14,17,19,21]. However, the significance of this observed phenomena with 3-"up" conformation in all particles of the Fab-S complex, in another word, its correlation to the neutralization advantages, remains unknown.

Interestingly, we noted that 13G9 and 58G6, though originally isolated from the samples of different COVID-19 convalescent donors, were both transcribed from IGHV1–58 and IGKV3–20. These 2 variant regions were also genetically responsible for a panel of reported neutralizing Abs with high potency against

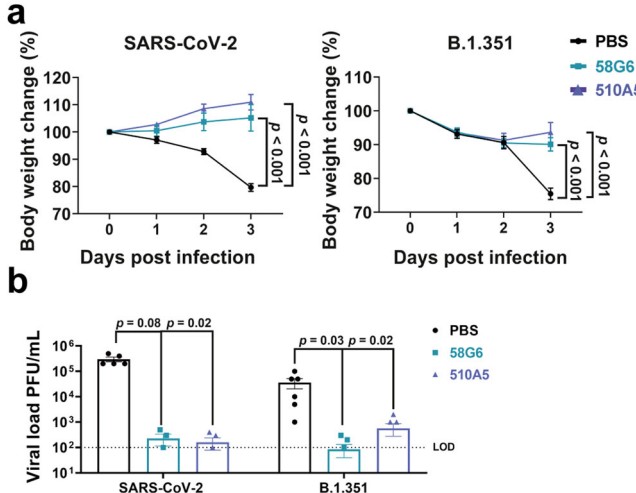

**Fig. 6 The prophylactic efficacy of 58G6 or 510A5 in hACE2 transgenic mice challenged with authentic viruses. a** Body weight changes were recorded for PBS (SARS-CoV-2 (WIV04): $n = 5$; B.1.351: $n = 6$), 58G6 (SARS-CoV-2 (WIV04): $n = 4$; B.1.351: $n = 7$) and 510A5 (SARS-CoV-2 (WIV04): $n = 5$; B.1.351: $n = 7$) treatment groups. All the mice received one dose of antibodies (10 mg/kg body weight) injected (i.p.) 24 h prior to the intranasal challenge with SARS-CoV-2 (WIV04) (left) or B.1.351 (right). Equal volume of PBS was used as negative control. The weight loss was recorded over a course of 3 days. **b** The virus loads in infected lungs of the PBS group, the 58G6 group and the 13G9 group as shown in **a** were determined by PRNT at 3 days post infection (dpi). The mean values ± SEM of all data points were shown. $p$ values were calculated via two-sided Student's $t$-test. Dashed line indicated assay limits of detection (LOD).

SARS-CoV-2 as well as B.1.351[5,6,20,21,27]. These findings highlighted the otherwise overlooked importance for the pairing of the IGHV1–58 and the IGKV3–20 germline genes in neutralizing SARS-CoV-2 and its variant.

In conclusion, we present 2 potent SARS-CoV-2 RBD specific mAbs with exceptional efficacy against B.1.351, for which a significant proportion of reported neutralizing Abs are impaired. Structural analysis of epitopes revealed the potential neutralizing mechanism of neutralizing Abs against B.1.351 carrying the E484K mutation. These broad-spectrum neutralizing Abs could be promising candidates for the prophylaxis and therapeutic interventions of the pandemic of SARS-CoV-2 variants carrying escape mutations.

## Methods
**Patient information and Isolation of antibodies to SARS-CoV-2**. We have complied with all relevant ethical regulations for work with human participants, and that informed consent was obtained. This study has been approved by the Ethics Board of Chongqing Medical University.

The 74 mAbs analyzed in our manuscript were derived from a total of 39 COVID-19 convalescent blood samples collected within a 2-month window post discharge. These 39 convalescent patients have an average age of 45 years old, and majority of them exhibited mild symptoms. Briefly, we utilized SARS-CoV-2 RBD as bait to sort a series of pooled antigen-specific memory B cells from 5 to 8 COVID-19 convalescent patients. The IgG heavy and light chains of mAbs genes in these memory B cells were obtained by single cell PCR and transiently transfected into Lenti-X293T cells (TAKARA, 632180) for the identification of mAbs with capabilities of the neutralization against SARS-CoV-2 pseudovirus. With such rapid screening system, we were capable to obtain the defined neutralizing Abs within 6 days.

**Recombinant antibody production and purification**. A pair of plasmids separately expressing the heavy- and the light- chain of antibodies were transiently co-transfected into Expi293™ cells (ThermoFisher, A14528) with ExpiFectamine™ 293 Reagent. Then the cells were cultured in shaker incubator at 120 rpm and 8% $CO_2$ at 37 °C. After 7 days, the supernatants with the secretion of antibodies were collected and captured by protein G Sepharose (GE Healthcare). The bound

antibodies on the Sepharose were eluted and dialyzed into phosphate-buffered saline (PBS). The purified antibodies were used in following binding and neutralization analyses.

**Authentic SARS-CoV-2 neutralization assay**. The neutralizing potency of mAbs against authentic SARS-CoV-2 virus quantified via qRT-PCR was performed in a biosafety level 3 laboratory of Fudan University. Serially diluted mAbs or mAb mixture (1:1 with same quality) were incubated with authentic SARS-CoV-2 virus (nCoV-SH01, GenBank: MT121215.1, 100 TCID50) for 1 h at 37 °C. After the incubation, the mixtures were then transferred into 96-well plates, which were seeded with Vero E6 cells. The plates were kept at 37 °C for 48 h. And the supernatant viral RNA load of each well was quantified by qRT-PCR. For qRT-PCR, the viral RNA was extracted from the collected supernatant using Trizol LS (Invitrogen) and used as templates for the qRT-PCR analysis by Verso 1-Step qRT-PCR Kit (Thermo Scientific) following the manufacturer's instructions. qRT-PCR was performed using the LightCycler 480 II PCR System (Roche) with the following program: 50 °C 15 mins; 95 °C 15 mins; 40 cycles of 95 °C 15 s, 50 °C 30 s, 72 °C 30 s. The sequences of primers are shown in Supplementary Table 3.

The neutralizing potency of mAbs against authentic SARS-CoV-2 and B.1.351 viruses was performed via PRNT in a biosafety level 3 laboratory of Wuhan Institute of Virology. Each mAb sample was serially diluted with DMEM as two folds and the sample quality, mixed with equal volume of authentic SARS-CoV-2 virus (WIV04, GenBank: MN996528.1) or SARS-CoV-2 South Africa strain B.1.351 (NPRC 2.062100001, GenBank: MW789246.1) and incubated at 37 °C for 1 h. Vero E6 cells in 24-well plates were inoculated with the antibody-virus mixture at 37 °C, 1 h. Later, the mixture was replaced with DMEM containing 2.5% FBS and 0.8% carboxymethylcellulose. The plates were fixed with 8% paraformaldehyde and stained with 0.5% crystal violet 4 days later. All samples were tested in duplicate and neutralization titers were defined as the serum dilution resulting in a plaque reduction of at least 50%[28].

**Sequence analysis of antigen-specific mAbs**. IMGT/V-QUEST (http://www.imgt.org/ IMGT_vquest /vquest) and IgBLAST (https://www.ncbi.nlm.nih.gov/igblast/), MIXCR (https://mixcr.readthedocs.io/en/master/) and VDJtools (https://vdjtools-doc.readthedocs.io/en/master/overlap.html) tools were used to do the variable region analysis and annotation for each antibody clone.

**Production of pseudovirus bearing S protein**. pVSVG expressing SARS-CoV-2 S protein was constructed as using the packaging plasmid (VSV-G pseudotyped ΔG-luciferase)[29]. It encoded either the S protein of SARS-CoV-2, B.1.1.7 or chimeric construct including B.1.351 RBD and $S^{D614G}$ was generated. Lenti-X293T cells were grown to 80% confluency before transfection with VSV-G pseudotyped ΔG-luciferase, pWPXL and pSFAX2. These cells were cultured overnight at 37 °C with 5% $CO_2$. DMEM supplemented with 5% fetal bovine serum and 100 IU/mL of penicillin and 100 μg/mL of streptomycin was added to the inoculated cells, which were cultured overnight for 72 h. The supernatant was harvested, filtered by 0.45 μm filter and centrifuged at 300 g for 10 mins to collect the supernatant, then aliquoted and storied at −80 °C.

**Pseudovirus neutralization assay**. Serially diluted mAbs with volume of 50 μL were incubated with the same volume of the Lenti-X293T cell supernatants containing pseudovirus for 1 h at 37 °C. These pseudovirus-antibody mixtures were added to ACE2 expressing Lenti-X293 T cells (293 T/ACE2). After 72 h, the luciferase activities of infected 293T/ACE2 cells were detected by the Bright-Luciferase Reporter Assay System (Promega, E2650). The $IC_{50}$ and $IC_{80}$ of the evaluated mAbs were tested by the Varioskan LUX Microplate Spectrophotometer (Thermo Fisher), and calculated by a four-parameter logistic regression using GraphPad Prism 8.0.

**S Protein expression and purification**. To express the prefusion S ectodomain, the gene encoding residues 1-1208 of SARS-CoV-2 S (GenBank: MN908947.3) with a C-terminal T4 fibritin trimerization motif, an HRV-3C protease cleavage site, a Twin-Strep-tag and an 8 × His-tag was synthesized, and cloned into the mammalian expression vector pcDNA3.1, which was a kind gift from L. Sun at Fudan University, China. The gene of the S protein was constructed with proline substitutions at residues 986 and 987, a "GSAS" instead of "RRAR" at the furin cleavage site (residues 682–685) according to Jason S. McLellan's research[25].

Expi293 cells (Thermo Fisher Scientific, USA) cultured in Freestyle 293 Expression Medium (Thermo Fisher Scientific, USA) were maintained at 37 °C. Cells were diluted to a density of $2.5 \times 10^6$ to $3 \times 10^6$ cells per mL before transfection. For protein production, 1.2 mg DNA was mixed with 3 mg polyethyleneimine in 30 mL Freestyle 293 Expression Medium, incubated for 20 mins, then added to 1000 mL of cells[30]. Transfected cells were cultured at 35 °C, and the cell culture supernatant was collected at day 4 to day 5.

S protein was purified from filtered cell supernatants using Strep-Tactin resin (IBA) before being subjected to additional purification by gel filtration chromatography using a Superose 6 10/300 column (GE Healthcare, USA) in 1 × PBS, pH 7.4 (Supplementary Fig. 10a, b).

**Cryo-EM sample preparation and data collection.** Purified SARS-CoV-2 S was diluted to a concentration of 1.5 mg/mL in PBS, pH 7.4. 5 μL of purified SARS-CoV-2 S was mixed with 1 μL of 58G6 Fab fragments at 2 mg/mL in PBS and incubated for 30 mins on ice. A 3 μL aliquot of the mixture (added with 0.01% DDM) was applied onto an $H_2/O_2$ glow-discharged, 300-mesh Quantifoil R1.2/1.3 grid (Quantifoil, Micro Tools GmbH, Germany). The grid was then blotted for 3.0 s with a blot force of −1 at 8 °C and 100% humidity and plunge-frozen in liquid ethane using a Vitrobot (Thermo Fisher Scientific, USA). Cryo-EM data sets were collected at a 300 kV Titan Krios microscope (Thermo Fisher Scientific, USA) equipped with a K3 detector (Gatan, USA). The exposure time was set to 2.4 s with a total accumulated dose of 60 electrons per $Å^2$, which yields a final pixel size of 0.82 Å. 2605 micrographs were collected in a single session with a defocus range comprised between 1.0 and 2.8 μm using SerialEM. The sample preparation and data collection for the SARS-CoV-2 S-13G9 Fab complex were in accordance with the SARS-CoV-2 S-58G6 Fab complex. The statistics of cryo-EM data collection can be found in Supplementary Table 1.

**Cryo-EM data processing.** All dose-fractioned images were motion-corrected and dose-weighted by MotionCorr2 software[31] and their contrast transfer functions were estimated by cryoSPARC patch CTF estimation[32]. For the dataset of SARS-CoV-2 S-58G6 Fab complex, a total of 1,255,599 particles were auto-picked using the template picker and 820,872 raw particles were extracted with a box size of 512 pixels in cryoSPARC[32]. The following 2D, 3D classifications, and refinements were all performed in cryoSPARC. 237,062 particles were selected after two rounds of 2D classification, and these particles were used to do Ab-Initio reconstruction in six classes. Then these six classes were used as 3D volume templates for heterogeneous refinement with all selected particles, with 108,020 particles converged into the SARS-CoV-2 S-58G6 Fab class. Next, this particle set was used to perform non-uniform refinement, yielding a resolution of 3.56 Å.

For the dataset of SARS-CoV-2 S-13G9 Fab complex, a total of 445,137 particles were auto-picked using the template picker and 266,357 raw particles were extracted with a box size of 512 pixels in cryoSPARC. The following 2D, 3D classifications, and refinements were all performed in room temperature (RT). 70,519 particles were selected after two rounds of 2D classification, and these particles were used to do Ab-Initio reconstruction in six classes. Then these 6 classes were used as 3D volume templates for heterogeneous refinement with all selected particles, with 52,880 particles converged into the SARS-CoV-2 S-13G9 Fab class. Next, this particle set was used to perform non-uniform refinement, yielding a resolution of 3.92 Å.

Although the overall resolution for these structures is up to 3.5 Å - 3.6 Å for 58G6 and 3.9 Å - 4.0 Å for 13G9, the maps for the binding interface between RBD and Fabs are quite weak due to the conformational heterogeneity of the RBD, which is similar to previous structural investigations[14,17,21,33]. To improve the resolution for the binding interface, we subsequently added local refinement processing. A local reconstruction focusing on the RBD-Fabs region was carried out. Furthermore, the density map for the binding interface could be improved further by local averaging of the RBD-Fab equivalent copies, finally yielding a 3.5 Å map of the region corresponding to the 58G6 variable domains and RBD (Supplementary Fig. 10g, j). Similarly, we improve the local resolution between 13G9 variable domains and RBD up to 3.8 Å (Extended Fata Fig. 10c, f).

Local resolution estimation, filtering, and sharpening were also carried out using cryoSPARC. The full cryo-EM data processing workflow is described in Supplementary Fig. 10 and the model refinement statistics can be found in Supplementary Table 1.

**Model building and refinement.** To build the structures of the SARS-CoV-2 S-58G6 Fab and S-13G9 Fab complexes, the structure of the SARS-CoV-2 S gly-coprotein in complex with the C105 neutralizing antibody Fab fragment[14] (PDB: 6XCN) was placed and rigid-body fitted into the cryo-EM electron density maps using UCSF Chimera[34], respectively. Both of the 58G6 and 13G9 Fab models were first predicted using Phyre2[26] and then manually built in Coot 0.9[35] with the guidance of the cryo-EM electron density maps, and overall real-space refinements were performed in Phenix 1.18[36]. The data validation statistics are shown in Supplementary Table 1. The corresponding structure data for the 13G9/SARS-CoV-2 RBD complex and 58G6/SARS-CoV-2 RBD complex was available in wwPDB as 7E3K (https://doi.org/10.2210/pdb7E3K/pdb) and 7E3L (https://doi.org/10.2210/pdb7E3L/pdb) respectively.

**Creation of figures.** Figures of molecular structures were generated using PyMOL[37] and UCSF ChimeraX[38].

**Surface plasmon resonance (SPR) experiments.** The affinity of the neutralizing Abs binding to the S1 subunit of SARS-CoV-2 or B.1.351 was measured using the Biacore X100 platform at RT. A CM5 chip (GE Healthcare) was linked with anti-human IgG-Fc antibody (Cytiva, BR-1008-39) to capture about 9000 response units of the neutralizing Abs. The gradient concentrations of SARS-CoV-2 S1 or an artificial chimeric construct carrying 3 mutations on B.1.351 RBD and $S^{D614G}$ (B.1.351 S1) (Sino Biological, Beijing, China) were prepared (2-fold dilutions, from 50 nM to 0.78 nM) with HBS-EP+ Buffer (0.01 M HEPES, 0.15 M NaCl, 0.003 M

EDTA and 0.05% (v/v) Surfactant P20, pH 7.4), and sequentially injected into the chip and monitored for the binding kinetics. After the final reading, the sensor surface of the chip was regenerated with 3 M $MgCl_2$ (GE) before the measurement of the next mAb. The affinity was calculated with Biacore X100 Evaluation Software (Version:2.0.2) using 1:1 binding fit model.

To determine competition with the ACE2 peptidase domain, SARS-CoV-2 RBD was coated on a CM5 sensor chip via amine group for a final RU around 250. The top 20 neutralizing Abs (20 μg/mL) were injected onto the chip until binding steady-state was reached. ACE2 (20 μg/mL) was then injected for 60 s. Blocking efficacy was determined by comparison of response units with and without prior antibody incubation.

**Competitive ELISA.** For competitive ELISA used in epitope mapping of mAbs, 2 μg/mL recombinant RBD-his (Sino Biological, Beijing, China) was added in 384-well plates and incubated at 4 °C overnight. 50 μg/mL mAbs per well were added. The plates were incubated at 37 °C for 1 h and then washed. Biotinylation of mAbs (the top 20 neutralizing Abs and 81A11, previously reported SARS-CoV CR3022[24]) were performed using the EZ-link NHS-PEO Solid Phase Biotinylation Kit (Pierce) according to the manufacturer's protocol and purified using MINI Dialysis Unit (ThermoFisher, 69576). 500 ng/mL biotinylated mAbs were added to each well, and the plates were incubated at 37 °C for 1 h. ALP-conjugated streptavidin (Mabtech, Sweden, 3310-10) was added at 1:1000, followed by an incubation of 30 mins at 37 °C. For the quantification of bound IgG, PNPP (Thermo Fisher) was added at 1 mg/mL and the absorbance at 405 nm was measured by the MultiSkan GO fluoro-microplate reader (Thermo Fisher).

**Western blot analysis.** The recombinant RBD protein was mixed with 5 × loading buffer (Beyotime, Shanghai, China) and denatured for 5 mins at 100 °C. Denatured proteins (200 ng) were subjected to electrophoresis with 10% SDS-polyacrylamide gel and then transferred to PVDF membranes. After blocking by skim milk (Biofroxx), the membranes were incubated at 4 °C overnight, with the purified mAbs as primary Abs. The next days, the membranes were washed with TBST and incubated with HRP-conjugated mouse-anti-human Fc antibody (Abcam, ab99759, 1:10000) for 1 h at RT. The membranes were examined on ChemiDoc Imaging System (Bio-rad). The corresponding uncropped figures were included in the Supplementary Information.

**Peptide ELISA.** Peptide ELISA was performed with synthesized peptides overlapping with 5 amino acids (Genescripts, Wuhan, China). These peptides were tethered by N-terminal biotinylated linker peptides (biotin-ahx), except for the first peptide at the N-terminus, whose biotin was linked to the C terminus instead. The RBD9-1 amino acid residues were selected and mutated to alanine and synthesized by Genescripts (Wuhan, China). 50 μL synthesized peptide was added to the streptavidin-coated 384-well plate in duplets to make a final concentration of 5 μg/mL or other indicated concentrations. The plates were incubated for 2 h at RT. After washing, the plates were blocked with Protein-Free Blocking Buffer (Pierce, USA, 37573) at RT for 1 h and incubated with testing mAbs (10 μg/mL), ACE2 (10 μg/mL) or 58G6 with indicated concentrations at RT for another 1 h. Reacted mAbs were detected using ALP-conjugated Goat F(ab')₂ Anti-Human (IgG (Fab')₂) secondary antibody (Abcam, ab98532, 1:2000) for 30 mins at RT, followed with quantification detection.

For the ACE2 competitive peptide ELISA, 5 μg/mL synthesized RBD9-1 was immobilized on the streptavidin-coated 384-well plate at RT for 2 h. After washing with Protein-Free Blocking Buffer, the plates were blocked with this blocking buffer. Next, serial diluted 58G6 (20-0.625 μg/mL) in 50 μL of the blocking buffer were added into plate and the plates were incubated at RT for 1 h. Then, the plate incubated with 2 μg/mL ACE2 at RT for another 1 h. The ELISA plates were washed 4 times by blocking buffer and 50 μL Goat F(ab')₂ Anti-Human (IgG (Fab')₂) secondary antibody conjugated with ALP (Abcam, ab98532, 1:2000) was incubated with the plate at RT for 30 mins. The plate was washed and followed with quantification detection.

**Authentic SARS-CoV-2 and B.1.351 viruses and animal study.** We have complied with all relevant ethical regulations for animal testing and research. All the mice in this study were cared following the National Institutes of Health Guidelines for the Care and Use of Experimental Animals. Viral infections were conducted in an animal biosafety level 3 (ABSL-3) facility at Wuhan Institute of Virology, with a protocol approved by the Laboratory Animal Ethics Committee of Wuhan Institute of Virology, Chinese Academy of Sciences (Permit number: WIVA26201701).

Protection efficacy of potent antibodies 58G6 and 510A5 was evaluated in an established hACE2 mouse model of SARS-CoV-2 infection. Six- to eight-week-old male human transgenic ACE2 mice were purchased from Gempharmatech (T037657). Animals were housed in groups of up to 5 mice/cage at 18–24 °C ambient temperature and 40–60% humidity. Mice were fed a 20% protein diet and mainted on a 12 h light/dark cycle. Food and water were available ad libtium.

These hACE2 mice were treated with 58G6 or 510A5 monoclonal antibody at a concentration of 10 mg/kg by intraperitoneal route, respectively. The mice treated with PBS were used as the negative control. 24 h later, all mice were intranasally infected with $10^5$ PFU authentic SARS-CoV-2 or B.1.351 viruses in a total volume

of 50 μL. At 3 days post infection of SARS-CoV-2 or B.1.351, the lungs of mice were collected for viral load determination using plaque assay[39].

**Data analysis**. Data are shown as mean ± SEM. Two-group comparisons were performed by Student's *t*-test. The difference was considered significant if $p < 0.05$.

**Reporting summary**. Further information on research design is available in the Nature Research Reporting Summary linked to this article.

## Data availability

Data generated and analyzed in this study are provided in the Source Data file. The coordinates and structure factor files for the 13G9/SARS-CoV-2 RBD complex and 58G6/SARS-CoV-2 RBD complex have been deposited in the Protein Data Bank (PDB) under accession number 7E3K (https://doi.org/10.2210/pdb7E3K/pdb) and 7E3L (https://doi.org/10.2210/pdb7E3L/pdb) respectively. All other data pertaining to this study are also available from the corresponding author upon reasonable requests. Antibody and antibody sequences are available (by contacting Aishun Jin from the Chongqing University Department of Immunology; aishunjin@cqmu.edu.cn) for research purposes only under an MTA, which allows the use of the antibody sequences for non-commercial purposes but not their disclosure to third parties. Source data are provided with this paper.

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

## Acknowledgements

This study was supported by the Emergency Project from Chongqing Medical University and Chongqing Medical University fund (X4457) with the donation from Mr. Yuling Feng. We acknowledge the clinical laboratories of Yongchuan Hospital of Chongqing Medical University and the Third Affiliated Hospital of Chongqing Medical University for providing blood samples. We acknowledge Dr. Xiaowu Chen from Stanford School of Medicine for helpful comments in the analysis of the structural data.We are grateful to Xiaoxiao Gao and Cheng Peng for their technical help, and to Wuhan National Biosafety Laboratory running team, including engineer, biosafety, biosecurity, and administrative staff. The SARS-CoV-2 South Africa strain (NPRC 2.062100001) was provided by Guangdong Provicial Center for Disease Control and Prevention. We also thank all healthy individuals participated in this study.

## Author contributions

A.J. and A.H. conceived and designed the study. F.L. and J.H. were responsible for antibody production and purification. J.W., K.W., J.H., S.L., N.T., G.Z. and Q.G. conducted the pseudovirus neutralization assays and Y.X., C.G., Y.W., W.X., X.C., D.Q. and Z.Y. performed authentic SARS-CoV-2 neutralization assays. S.L. and Y.H. played an import role in data analysis of neutralizing Abs sequences. T.L., Y.W., Y.L., S.S., Q.C., F.G. and M.S. performed ELISA, competitive ELISA and peptide ELISA. X.H., C.H., R.W. and S.M. were responsible for SPR assay for the affinity of these neutralizing Abs and competition of these neutralizing Abs with ACE2. H.G., F.L., Y.G., W.W., X.J. and H.Y. carried out the cryo-EM studies. H.Z., Y.Z., Z.Z., H.Z., N.L. and B.Z. were responsible for the prophylactic test of neutralizing Abs for hACE2 mice challenged with SARS-CoV-2 and B.1.351. L.L. and C.H. generated figures and tables, and take responsibility for the integrity and accuracy of the data presentation. A.J., T.L., W.W. and H.G. wrote the manuscript.

## Competing interests

The authors declare the following competing interests: Patent has been filed for some of the antibodies presented here (patent application number: PCT/CN2020/115480, PCT/CN2021/078150, PCT/CN2021/113261; patent applicants: Chongqing Medical University). A.J., T.L., X.H., Y.W., C.H., S.L., M.S. and J.W. are the inventors. All other authors declare no competing interests.
