## [Peer Review File · Nature Communications]

Peer Review Comments, initial response to Manuscript:

Reviewer #1 (Remarks to the Author):

In this revision of "Ultrapotent SARS-CoV-2 neutralizing antibodies with protective efficacy against newly emerged mutational variants", the authors have thoughtfully and thoroughly responded to the concerns and questions raised by this reviewer. Collectively, these data and alterations of the text have strengthened the manuscript; and also improved its clarity. It is a bit unfortunate that the mAbs do not cross-react to SARS-CoV (and, as the authors point out, due to sequence similarity to WIV1 are unlikely to bind/neutralize that virus). One minor suggestion is that with the improved structural data, it may be useful to include a sentence or two describing the structural explanation of why there is a lack of cross-reactivity (Perhaps including a sequence alignment of the various CoVs contacted by the mAbs described would help convey this point).

Reviewer #2 (Remarks to the Author):

The authors have addressed the comments raised by the reviewers and the paper is improved

Reviewer #3 (Remarks to the Author):

The authors appropriately addressed my concerns and expanded on their previous findings, which makes this revised manuscript a more complete and informative story. Authentic virus neutralization assays were performed against B.1.351 variant in direct comparison to WT SARS-CoV-2. The in vivo prophylactic efficacy of the two most potent mAbs was shown by testing them in the transgenic mouse model harboring human ACE2. The authors followed reviewers suggestions to provide a more refined model of specific interactions between the Fabs and RBD, which resulted in a valuable discussion on the role of 484, a position where mutations in VOC caused significant resistance to mAb neutralization potency. Interestingly, mAb 13G9 was found to be in direct contact with this residue, which would suggest a loss of neutralization potency against the E484K bearing variants, but instead this mAb still potently neutralizes authentic B.1.351. Additionally, the significance of the finding that the 2 described mAbs bind to an exclusively 3 "up" RBD conformation is unknown, as the authors state in the revised manuscript, but it is nevertheless informative, and this property could make these mAbs a potentially useful tool and subject for more detailed binding mechanism studies. Binding affinity studies of the mAbs to the RBD were included, as well as an experiment addressing cross-reactivity with SARS-CoV and MERS-CoV-2. Overall, the authors addressed major and minor issues brought up by other reviewers and myself, and adjusted the text of the manuscript accordingly.

There are still a few minor issues:

The neutralization curves of the B.1.351 virus do not always show nice sigmoidal curves and therefore the given IC50s are not very trustworthy for some of the antibodies compared to the IC50s of the WT virus. This makes comparing WT vs B.1.351 difficult. For example 58G6 second dilution has lower neutralization compared to 3rd dilution, shifting the curve to be more sensitive while all the other dilutions provide a very similar neutralization for WT and B.1.351. For 510A4, 51D3 and 81E1

for example the slope of the curve is very shallow making interpretation of the curve difficult. In addition, the authors reanalyzed the neutralization data in Fig 2b-d and by changing to IC80s the slight difference in favor of the cocktail is visible. However, they still mention the IC50s in the text (line 143). This could be changed or the IC50s could also be added to the figure. When discussing the synergistic effects of some of the mAbs from different clusters in lines 139-144, the conclusions are conflicting, but referencing the same figure- the antibodies show an advantage when mixed as a cocktail, but are also potent enough on their own. This could be phrased better. In Fig 2b-d it would be useful to add from which group the antibodies are (which epitope they target), or otherwise label the mAbs selected for testing in cocktails in colors for better readability (Fig 2a).

The potency of the plasma of the patients from which the mAbs were isolated, in comparison to the monoclonal antibody response was not discussed in the revised manuscript.

In the text describing the in vivo study data, the author do not mention the direct comparison of WT and B.1.351 when looking at weight loss, the author just mention that there is protection in both, however there is indeed weight loss up to day 2 for B.1.351 while for WT no weight loss observed at any time. This could indicate some infection for B.1.351 and not for WT, which is a difference that should be addressed in the text.

When presenting the binding data in Extended Data Fig. 3, the scales on the y-axis (RU) should be the same when comparing one antibody against WT and B.1.3.51 S1. Mention in the figure legend that this was done by SPR.

The graph in Extended data Fig.1 could be improved some more- different/more colors, separate to several graphs, and the IC50 values should be presented in black.

In the legend of Extended data Fig.11 it is stated that the color scheme is the same as in Fig.3, but I would say it is like Fig. 4.or 5. Also, in the replies to the reviewers authors in multiple places refer to lines incorrectly which made it hard to tell what they adjusted in the manuscript. This might be a Word mistake, but should be checked more carefully.

REVIEWER COMMENTS

Reviewer #1 (Remarks to the Author):

In this revision of "Ultrapotent SARS-CoV-2 neutralizing antibodies with protective efficacy against newly emerged mutational variants", the authors have thoughtfully and thoroughly responded to the concerns and questions raised by this reviewer. Collectively, these data and alterations of the text have strengthened the manuscript; and also improved its clarity. It is a bit unfortunate that the mAbs do not cross-react to SARS-CoV (and, as the authors point out, due to sequence similarity to WIV1 are unlikely to bind/neutralize that virus). One minor suggestion is that with the improved structural data, it may be useful to include a sentence or two describing the structural explanation of why there is a lack of cross-reactivity (Perhaps including a sequence alignment of the various CoVs contacted by the mAbs described would help convey this point).

Reviewer #2 (Remarks to the Author):

The authors have addressed the comments raised by the reviewers and the paper is improved

Reviewer #3 (Remarks to the Author):

The authors appropriately addressed my concerns and expanded on their previous findings, which makes this revised manuscript a more complete and informative story. Authentic virus neutralization assays were performed against B.1.351 variant in direct comparison to WT SARS-CoV-2. The in vivo prophylactic efficacy of the two most potent mAbs was shown by testing them in the transgenic mouse model harboring human ACE2. The authors followed reviewers suggestions to provide a more refined model of specific interactions between the Fabs and RBD, which resulted in a valuable discussion on the role of 484, a position where mutations in VOC caused significant resistance to mAb neutralization potency. Interestingly, mAb 13G9 was found to be in direct contact with this residue, which would suggest a loss of neutralization potency against the E484K bearing variants, but instead this mAb still potently neutralizes authentic B.1.351. Additionally, the significance of the finding that the 2 described mAbs bind to an exclusively 3 "up" RBD conformation is unknown, as the authors state in the revised manuscript, but it is nevertheless informative, and this property could make these mAbs a potentially useful tool and subject for more detailed binding mechanism studies. Binding affinity studies of the mAbs to the RBD were included, as well as an experiment addressing cross-reactivity with SARS-CoV and MERS-CoV-2. Overall, the authors addressed major and minor issues brought up by other reviewers and myself, and adjusted the text of the manuscript accordingly.

There are still a few minor issues:

The neutralization curves of the B.1.351 virus do not always show nice sigmoidal curves and therefore the given IC50s are not very trustworthy for some of the antibodies compared to the IC50s of the WT virus. This makes comparing WT vs B.1.351 difficult. For example 58G6 second dilution has lower neutralization compared to 3rd dilution, shifting the curve to be more sensitive while all the other dilutions provide a very similar neutralization for WT and B.1.351. For 510A4, 51D3 and 81E1 for example the slope of the curve is very shallow making interpretation of the curve difficult.

In addition, the authors reanalyzed the neutralization data in Fig 2b-d and by changing to IC80s the slight difference in favor of the cocktail is visible. However, they still mention the IC50s in the text (line 143). This could be changed or the IC50s could also be added to the figure. When discussing the synergistic effects of some of the mAbs from different clusters in lines 139-144, the conclusions are conflicting, but referencing the same figure- the antibodies show an advantage when mixed as a cocktail, but are also potent enough on their own. This could be phrased better. In Fig 2b-d it would be useful to add from which group the antibodies are (which epitope they target), or otherwise label the mAbs selected for testing in cocktails in colors for better readability (Fig 2a).

The potency of the plasma of the patients from which the mAbs were isolated, in comparison to the monoclonal antibody response was not discussed in the revised manuscript.

In the text describing the in vivo study data, the author do not mention the direct comparison of WT and B.1.351 when looking at weight loss, the author just mention that there is protection in both, however there is indeed weight loss up to day 2 for B.1.351 while for WT no weight loss observed at any time. This could indicate some infection for B.1.351 and not for WT, which is a difference that should be addressed in the text.

When presenting the binding data in Extended Data Fig. 3, the scales on the y-axis (RU) should be the same when comparing one antibody against WT and B.1.351 S1. Mention in the figure legend that this was done by SPR.

The graph in Extended data Fig.1 could be improved some more- different/more colors, separate to several graphs, and the IC50 values should be presented in black.

In the legend of Extended data Fig.11 it is stated that the color scheme is the same as in Fig.3, but I would say it is like Fig. 4.or 5. Also, in the replies to the reviewers authors in multiple places refer to lines incorrectly which made it hard to tell what they adjusted in the manuscript. This might be a Word mistake, but should be checked more carefully.

REVIEWER COMMENTS

Reviewer #1 (Remarks to the Author):

In this revision of "Ultrapotent SARS-CoV-2 neutralizing antibodies with protective efficacy against newly emerged mutational variants", the authors have thoughtfully and thoroughly responded to the concerns and questions raised by this reviewer. Collectively, these data and alterations of the text have strengthened the manuscript; and also improved its clarity. It is a bit unfortunate that the mAbs do not cross-react to SARS-CoV (and, as the authors point out, due to sequence similarity to WIV1 are unlikely to bind/neutralize that virus). One minor suggestion is that with the improved structural data, it may be useful to include a sentence or two describing the structural explanation of why there is a lack of cross-reactivity (Perhaps including a sequence alignment of the various CoVs contacted by the mAbs described would help convey this point).

We thank the Reviewer for the encouraging comment and this constructive suggestion. Detailed comparison including a sequence alignment of the mAbs binding regions of the various CoVs was added and discussed. Corresponding information was shown in the newly added Extended Data Fig.4b and at lines 282-287 within the text file.

Reviewer #2 (Remarks to the Author):

The authors have addressed the comments raised by the reviewers and the paper is improved

We thank the Reviewer for this positive comment.

Reviewer #3 (Remarks to the Author):

The authors appropriately addressed my concerns and expanded on their previous findings, which makes this revised manuscript a more complete and informative story. Authentic virus neutralization assays were performed against B.1.351 variant in direct comparison to WT SARS-CoV-2. The in vivo prophylactic efficacy of the two most potent mAbs was shown by testing them in the transgenic mouse model harboring human ACE2. The authors followed reviewers suggestions to provide a more refined model of specific interactions between the Fabs and RBD, which resulted in a valuable discussion on the role of 484, a position where mutations in VOC caused significant resistance to mAb neutralization potency. Interestingly, mAb 13G9 was found to be in direct contact with this residue, which would suggest a loss of neutralization potency against the E484K bearing variants, but instead this mAb still potently neutralizes authentic B.1.351. Additionally, the significance of the finding that the 2 described mAbs bind to an exclusively 3 "up" RBD conformation is unknown, as the authors state in the revised manuscript, but it is nevertheless informative, and this property could make these mAbs a potentially useful tool and subject for more detailed binding mechanism studies. Binding affinity studies of

the mAbs to the RBD were included, as well as an experiment addressing cross-reactivity with SARS-CoV and MERS-CoV-2. Overall, the authors addressed major and minor issues brought up by other reviewers and myself, and adjusted the text of the manuscript accordingly.

We are grateful for these constructive comments.

There are still a few minor issues:

The neutralization curves of the B.1.351 virus do not always show nice sigmoidal curves and therefore the given IC₅₀s are not very trustworthy for some of the antibodies compared to the IC₅₀s of the WT virus. This makes comparing WT vs B.1.351 difficult. For example 58G6 second dilution has lower neutralization compared to 3rd dilution, shifting the curve to be more sensitive while all the other dilutions provide a very similar neutralization for WT and B.1.351. For 510A4, 51D3 and 81E1 for example the slope of the curve is very shallow making interpretation of the curve difficult.

We thank the Reviewer for pointing out this point. Indeed, the sigmoidal curves have presented an issue to determine the precise IC₅₀s for some of our mAbs during the course of our study. Therefore, we drew a careful conclusion that 58G6 exhibited neutralizing potency against both the WT and B.1.351 viruses, without detailed comparison of the difference in the neutralization curves. For the mAbs showing irregular curves of B.1.351, we acknowledged the loss of their neutralizing potency against this mutant virus in the result descriptions and removed the description about IC₅₀ of 58G6 and 510A5 against B.1.351 at lines 93-94 in the updated manuscript and the IC₅₀ dash lines in the corresponding images (please see revised Figure 1).

In addition, the authors reanalyzed the neutralization data in Fig 2b-d and by changing to IC₈₀s the slight difference in favor of the cocktail is visible. However, they still mention the IC₅₀s in the text (line 143). This could be changed or the IC₅₀s could also be added to the figure. When discussing the synergistic effects of some of the mAbs from different clusters in lines 139-144, the conclusions are conflicting, but referencing the same figure- the antibodies show an advantage when mixed as a cocktail, but are also potent enough on their own. This could be phrased better. In Fig 2b-d it would be useful to add from which group the antibodies are (which epitope they target), or otherwise label the mAbs selected for testing in cocktails in colors for better readability (Fig 2a).

We thank the Reviewer for these suggestions. Accordingly, we removed the descriptions of the IC₅₀s from the synergistic study, and focused on the advances of the cocktail. Please see lines 130-132. Also, the specific groups for the antibodies used in the combination treatment were marked in the updated Fig. 2b-d.

The potency of the plasma of the patients from which the mAbs were isolated, in comparison to the monoclonal antibody response was not discussed in the revised manuscript.

We thank the Reviewer for this comment. The explanation and the discussion for the lack of this comparison have been added in the updated manuscript at lines 247-251 and lines 316-318.

In the text describing the *in vivo* study data, the author do not mention the direct comparison of WT and B.1.351 when looking at weight loss, the author just mention that there is protection in both, however there is indeed weight loss up to day 2 for B.1.351 while for WT no weight loss observed at any time. This could indicate some infection for B.1.351 and not for WT, which is a difference that should be addressed in the text.

We agree with the Reviewer that there are some differences that need to be addressed for the *in vivo* findings of the WT and B.1.351 infections. This is described in the updated manuscript at lines 223-225.

When presenting the binding data in Extended Data Fig. 3, the scales on the y-axis (RU) should be the same when comparing one antibody against WT and B.1.3.51 S1. Mention in the figure legend that this was done by SPR.

We thank the Reviewer for these constructive comments and have adjusted this figure with updated legend (see revised Extended Data Fig. 3).

The graph in Extended data Fig.1 could be improved some more- different/more colors, separate to several graphs, and the IC50 values should be presented in black.

We thank the Reviewer for this suggestion and have revised this figure.

In the legend of Extended data Fig.11 it is stated that the color scheme is the same as in Fig.3, but I would say it is like Fig. 4.or 5. Also, in the replies to the reviewers authors in multiple places refer to lines incorrectly which made it hard to tell what they adjusted in the manuscript. This might be a Word mistake, but should be checked more carefully.

We apologize for these mistakes and have made corresponding corrections throughout the revised manuscript.